# Effect of a Practice-Oriented Electronic Medical Record Education Program for New Nurses

**DOI:** 10.3390/healthcare13040365

**Published:** 2025-02-08

**Authors:** Jae-Kyun Ju, Hye-Won Jeong

**Affiliations:** 1Department of Surgery, Chonnam National University Hospital, Gwangju 61469, Republic of Korea; ge00023@cnuh.com; 2Department of Surgery, Chonnam National University Medical School, Gwangju 61469, Republic of Korea; 3Department of Nursing, Korea National University of Transportation, Jeungpyeong-gun 27909, Republic of Korea

**Keywords:** new nurses, electronic medical record, clinical competence, program evaluation

## Abstract

**Background/Objectives**: New nurses often face challenges in adapting to clinical environments, particularly in mastering electronic medical record (EMR) systems, which are critical for effective patient care and communication. This study aimed to evaluate the effectiveness of a practice-oriented EMR education program designed to improve new nurses’ EMR competencies. **Methods**: A quasi-experimental pretest–post-test design with a non-equivalent control group was employed. Fifty-four new nurses employed for less than a year participated, with 25 in the intervention group and 29 in the comparison group. The intervention group underwent five weekly sessions focused on core EMR tasks, including admission nursing, operation/procedure documentation, patient transfer/discharge, night duties, and SBAR handovers. The program, led by clinical nurse educators, incorporated lectures, practical exercises, and Q&A sessions. EMR competencies were assessed using a validated 5-point Likert scale. **Results**: The intervention group showed significant improvements across all assessed domains, with post-program scores significantly higher than those of the comparison group. The most notable improvements were in operation/procedure documentation and patient transfer/discharge tasks. The comparison group’s gains were limited, likely reflecting natural skill acquisition through clinical experience. **Conclusions**: The practice-oriented EMR education program effectively enhanced new nurses’ EMR competencies. The program’s structured approach, which combined theoretical instruction with extensive hands-on practice and department-specific adaptations, proved particularly effective in improving complex documentation tasks. The integration of comprehensive EMR training into nursing curricula and the expansion of such programs to other institutions are recommended for broader implementation.

## 1. Introduction

Since the advent of modern information technology and the subsequent introduction of Electronic Medical Record (EMR) systems, there has been a growing emphasis on the importance of utilizing the EMRs [1]. These systems are crucial not only for delivering healthcare services but also for facilitating communication and problem-solving among medical professionals [2]. Korean healthcare institutions have consistently striven to establish and advance EMR systems. Currently, they are implementing next-generation Hospital Information Systems (HIS) that leverage cloud technology, thereby establishing a standardized framework for healthcare information management [3]. An EMR is a computerized system accessed and managed through a computer, enabling the input and comprehensive management of a patient’s overall health information, including medical history, medication prescriptions, vital signs, and various test results [4]. Beyond facilitating standardized prescriptions in healthcare service delivery, EMRs also play a critical role in managing knowledge systems within hospital settings [5].

The ability to effectively utilize EMR systems and digital technologies has been shown to positively influence the quality of nursing care and improve patient clinical outcomes [6]. Recent studies particularly highlight that EMR systems contribute to reducing medical errors, enhancing communication, and improving patient safety [7]. Consequently, all nurses are expected to acquire proficient EMR competencies [8]. In particular, since new nurses who have recently joined medical institutions will be required to use EMR systems in their clinical practice, it is essential to enhance their EMR utilization skills to support their adaptation to the departmental environment and enable them to perform standardized tasks [9]. However, in the early stages of employment, new nurses often struggle to navigate medical records and perform various complex EMR-related tasks [10].

EMR systems exert a significant influence on the workflow of nurses, who constitute the largest group among healthcare providers [11]. The evolution of EMR systems has led to an increase in EMR-related tasks, which, along with other attention-demanding responsibilities, contribute to heightened job stress among clinical nurses [12]. Job stress, which is closely associated with burnout, reduced job satisfaction, and decreased performance, ultimately hinders adaptation to the hospital environment, making active institutional support from healthcare organizations essential [13,14]. In particular, new nurses face significant challenges in adapting to clinical practice for up to one year after employment due to the gap between theoretical knowledge and the practical skills required in real-world settings [15]. This difficulty in adaptation often results in a high turnover rate, with the turnover rate for newly graduated nurses with less than one year of experience in South Korea reaching 57.4% in 2022, the highest among all nursing groups [16].

A systematic practical training program has been shown to enhance the clinical performance of new nurses and facilitate problem-solving skills [17]. Given its positive impact on problem-solving abilities, it is imperative to implement practice-oriented and periodic educational interventions to assist new nurses in adapting to clinical settings and transitioning into competent professionals [18]. In particular, to improve new nurses’ understanding of the EMR system and alleviate difficulties in clinical adaptation, an effective and standardized EMR education program with a strong emphasis on practical application is essential [19]. Since new nurses’ practical experiences and competency levels vary depending on their length of employment and the characteristics of their assigned departments [20,21], a differentiated curriculum tailored to these factors is necessary.

A review of previous EMR-related educational studies reveals that while some research has focused on nursing programs designed to enhance EMR-related competencies, including the integration of academic EMR training into undergraduate nursing education, the majority of studies have primarily investigated nurses’ awareness of and satisfaction with EMR system utilization [8,9,22,23]. An Australian study on nurses using an EMR system reported that due to the lack of regular EMR education programs, nurses had to study on their own to perform EMR-related tasks [11]. Similarly, there is currently no independent and standardized curriculum aimed at improving the EMR competencies of nurses in Korean medical institutions. In particular, since EMR systems have been newly implemented, systematic EMR training for new nurses—who require training for successful adaptation—remains insufficient [24]. To address this issue, this study implemented a practice-oriented EMR education program (PEEP) led by clinical nurse educators. The program aimed to enhance the EMR competencies of new nurses and assess its effectiveness.

Building upon prior research and addressing these identified gaps, we formulated three research hypotheses. First, we hypothesized that new nurses who participated in the PEEP would demonstrate improved EMR practical abilities compared to their pre-program performance levels. Second, we anticipated that nurses who completed the PEEP would achieve higher EMR practical ability scores than those who did not participate in the program. Finally, we hypothesized that the magnitude of improvement in EMR practical abilities would be significantly greater in the intervention group than in the comparison group. By testing these hypotheses, we aimed to evaluate the effectiveness of a structured, practice-oriented approach to EMR education for new nurses.

## 2. Materials and Methods

### 2.1. Study Design

This study employed a pretest–post-test design with a non-equivalent control group to implement PEEP for new nurses and evaluate its effectiveness (Figure 1).

### 2.2. Participants

The study targeted nurses at Chonnam National University Hospital, located in Gwangju Metropolitan City, South Korea who met the following inclusion criteria: (1) employed for less than one year, (2) working full-time in direct patient care, (3) able to understand the purpose of the study and voluntarily agreeing to participate, and (4) no prior participation in a structured EMR education program. The required sample size was determined using the G*Power 3.1.9.7 program, referencing prior studies on new nurses [25]. With parameters set to a significance level of 0.05, power of 0.80, effect size of 0.80, and a two-tailed test for an independent sample *t*-test, the calculation indicated a need for 26 participants per group. Anticipating potential dropouts, we initially recruited 27 nurses for the intervention group and 29 nurses for the comparison group through convenience sampling. During the program period, two nurses in the intervention group were unable to attend the first session due to ward circumstances, resulting in final group sizes of 25 and 29 nurses in the intervention and comparison groups, respectively, with a total of 54 participants completing the study. A post hoc power analysis using G*Power confirmed that the achieved sample size provided a statistical power of 0.82, exceeding the conventional threshold of 0.80 for detecting meaningful differences between groups.

### 2.3. Research Procedure

#### 2.3.1. Development of the PEEP

The components of the PEEP were selected by one nursing education team leader and five clinical nurse educators with over 10 years of clinical nursing experience and Master’s degrees from C University Hospital to reflect the educational needs of new nurses in the previous year. The contents of the PEEP consisted of five sessions (admission nursing, operation/procedure nursing, department transfer/ward transfer/discharge nursing, night duty tasks and situation/background/assessment/recommendation (SBAR) and handover) that were essential for improving new nurses’ EMR competencies. The detailed educational content of each area was developed by nine clinical nurse educators from the nursing education team at C University Hospital who had more than seven years of clinical nursing experience. The completed program was evaluated using the content validity index (CVI) by two nursing department managers who were doctoral candidates, one clinical nurse educator with a PhD, and two nursing professors; the CVI was 0.95.

The PEEP was conducted over a five-week period, with five one-hour sessions. For admission nursing, practical exercises and Q&A sessions were held after lectures on workflow, patient information input, clinical observation records, dietary management, initial nursing information, home medication management, patient evaluation (falls, pressure ulcers, pain), and nursing notes. For operation and procedure nursing, practical exercises and Q&A sessions were conducted after lectures on electronic consent form review, confirmation of schedule, pre- and postoperative nursing records, procedure safety records, order fulfillment, treatment management, nursing note records, patient evaluation after operation and procedures (falls, pressure ulcers, pain), and records of nursing activities. For department transfer/ward transfer/discharge nursing, practical exercises and Q&A sessions were implemented after lectures on department transfer tasks, ward transfer tasks, prescription review and execution, document review and management, outpatient reservation management, discharge education, and discharge nursing records. For night duty tasks, practical exercises and Q&A sessions were conducted after lectures on next-day prescription management, medication and inspection barcode management, and product management and demand. For SBAR and handover, practical exercises and Q&A sessions were conducted after lectures on the importance of standardized communication, handover guidelines using SBAR, and identification of patient conditions before communication using an EMR system.

The PEEP was implemented in the hospital’s multimedia room to provide all new nurses with an opportunity to practice EMR work. One clinical nurse educator was assigned to three to four new nurses, and after a 10 min lecture on each area, practical exercises and Q&A sessions followed for 50 min. New nurses were allowed to freely ask questions during the practical exercises, and clinical nurse educators provided answers and guided their practice. Clinical nurse educators were assigned according to the new nurses’ work departments, and the program was designed so that educational exercises and Q&A sessions could be tailored to the characteristics of the departments. To facilitate practice and strengthen the EMR skills of new nurses, a checklist of essential items was created and distributed in each area, which the new nurses could reference upon returning to their work settings after completing the training.

#### 2.3.2. Application of PEEP

Data collection was conducted from 3 October 2022 to 30 January 2023, after obtaining approval from the nursing department of Chonnam National University Hospital. The PEEP consisted of five sessions, with each session held once a week for one hour, resulting in a total training period of five weeks. All sessions included a brief lecture by clinical nurse educators on the workflow of each area—“Admission Nursing”, “Operation/Procedure Nursing”, “Department Transfer/Ward Transfer/Discharge Nursing”, “Night Duty Tasks”, and “SBAR and Handover”—followed by hands-on practice and a Q&A session. A pre-survey was conducted one week before the start of the PEEP, and a post-survey was conducted one month after the end of the PEEP in each class to evaluate the EMR practical abilities of the intervention and comparison groups. To prevent cross-contamination between groups, the comparison group and the intervention group were not informed of each other’s membership, and the PEEP was conducted independently in the multimedia room, exclusively for the intervention group.

#### 2.3.3. Ethical Consideration

This study was conducted after receiving approval from the Institutional Review Board of Chonnam National University Hospital (IRB No: CNUH-2022-338). To ensure the voluntary participation of the study subjects, written informed consent was obtained after explaining the purpose and methodology of the study, the anticipated benefits, and any potential risks. Participants were informed that they could withdraw from the study at any time at their discretion. It was also explained that their anonymity would be guaranteed and that the research results would not be used for any purpose other than academic research. The study adhered to the ethical guidance set forth in the Declaration of Helsinki. Informed consent was obtained from all participating new nurses. For participants in the comparison group, the same PEEP provided to the intervention group was offered upon their request after the completion of data collection.

### 2.4. Measurements

To measure the EMR practical abilities of new nurses, nine clinical nurse educators at Chonnam University Hospital with over seven years of clinical experience developed the questions. The questionnaire consisted of 17 items each for admission nursing, operation/procedure nursing, and department transfer/ward transfer/discharge nursing, 10 items for night duty tasks, and 13 items for SBAR and handover. The questionnaire used a 5-point Likert scale ranging from “not at all proficient” (1 point) to “high proficient” (5 points). A higher score indicated greater EMR competencies in the respective area. This questionnaire was reviewed to determine the CVI by two nursing department managers who were doctoral candidates, one clinical nurse educator with a PhD, and two nursing professors. The total CVI was 0.97, and the CVI for each subarea was 0.98 for admission nursing, 0.96 for operation/procedure nursing, 1.0 for department transfer/ward transfer/discharge nursing, 0.96 for night duty tasks, and 0.97 for SBAR and handover. In this study, the total Cronbach’s α was 0.98, and Cronbach’s α for each subarea was 0.94 for admission nursing, 0.97 for operation/procedure nursing, 0.94 for department transfer/ward transfer/discharge nursing, 0.88 for night duty tasks, and 0.95 for SBAR and handover (Appendix A).

### 2.5. Data Analysis

The collected data were analyzed using the SPSS/WIN 26.0 (SPSS Inc., Chicago, IL, USA) program. Descriptive statistics were used to analyze whether the general characteristics and the EMR competencies data were normally distributed. Given the small sample size (fewer than 50 participants) in this study, normality was assessed using the Shapiro–Wilk test. For the homogeneity test for baseline participant characteristics, parametric and non-parametric statistical tests were conducted. An independent *t*-test or the Mann–Whitney U test was used for continuous data, while the χ^2^ test and Fisher’s exact test were applied to categorical data. To evaluate the effectiveness of the intervention, the differences in EMR competencies scores for each area were analyzed using a paired *t*-test or the Wilcoxon signed-rank test. The mean difference between pre-PEEP and post-PEEP scores in the intervention and comparison groups was analyzed using an independent *t*-test or the Mann–Whitney U test. For post hoc comparisons of post-PEEP variables between the intervention and comparison groups, when pre-test scores were non-homogeneous, ANCOVA was performed. The reliability of measurement variables was calculated using Cronbach’s α coefficient.

## 3. Results

### 3.1. General Characteristics of the Participants

The general characteristics of the study participants are shown in Table 1. The participants included 25 and 29 new nurses in the intervention and comparison groups, respectively. The general characteristics, such as age, gender, and working department were similar between the two groups; however, there was a significant difference in clinical experience (months) (χ^2^ = −3.60, *p* < 0.001). The pre-scores for EMR practical competencies in operation/procedure nursing, department transfer/ward transfer/discharge nursing, and night duty tasks were similar between the two groups. However, there was a significant difference in the pre-scores for admission nursing (t = −2.36, *p* = 0.022) and SBAR and handover (t = −2.21, *p* = 0.027).

### 3.2. Effects of the PEEP

Table 2 presents the results of evaluating the effectiveness of the PEEP for new nurses. The intervention group showed highly significant differences in all pre- and post-scores for admission nursing (t = −7.02, *p* < 0.001), operation/procedure nursing (t = −11.97, *p* < 0.001), department transfer/ward transfer/discharge nursing (t = −8.49, *p* < 0.001), night duty tasks (z = −4.20, *p* < 0.001), and SBAR and handover (z = −4.29, *p* < 0.001) in EMR competencies. In the comparison group, significant differences were observed in the pre- and post-scores for admission nursing (t = −2.35, *p* = 0.026), operation/procedure nursing (t = −3.72, *p* = 0.001), department transfer/ward transfer/discharge nursing (t = −4.75, *p* < 0.001), night duty tasks (z = −2.28, *p* = 0.023), and SBAR and handover (z = −3.44, *p* = 0.001) in EMR competencies. The mean difference in EMR competencies before and after the PEEP between the intervention group and the comparison group was statistically significant for admission nursing (z = −4.34, *p* < 0.001), operation/procedure nursing (t = −5.66, *p* < 0.001), department transfer/ward transfer/discharge nursing (z = −4.13, *p* < 0.001), night duty tasks (z = −2.68, *p* = 0.007), and SBAR and handover (z = −3.25, *p* = 0.001). The post-score difference between the intervention and comparison groups also showed statistically significant differences in admission nursing (F = −18.14, *p* < 0.001), operation/procedure nursing (z = −2.83, *p* < 0.001), department transfer/ward transfer/discharge nursing (z = −3.32, *p* = 0.001), night duty task (z = −3.30, *p* = 0.001), and SBAR and handover (F = 12.31, *p* < 0.001).

## 4. Discussion

This study implemented a five-week PEEP to improve field adaptation by enhancing the EMR competencies of new nurses and to assess its effects. The findings demonstrated significant improvements across all measured domains in the intervention group, supporting all three study hypotheses.

In examining the baseline characteristics, nurses in the intervention group had lower pre-scores across all five EMR competency areas compared to the comparison group. This difference likely arose from the comparison group’s longer average clinical experience (approximately three months), which provided them with more opportunities for direct EMR usage before the pre-evaluation. This finding aligns with previous research indicating that clinical exposure naturally enhances EMR proficiency through experiential learning [26]. Among the five EMR competency areas, both groups initially scored lower in operation/procedure nursing and department transfer/ward transfer/discharge nursing compared to other areas. This pattern likely reflects the limited opportunities for new nurses in their first year to perform these specialized procedures independently [27]. The lower scores in these areas particularly highlight the need for structured training, as these tasks often involve complex documentation requirements and interdepartmental coordination [28].

The significant improvement in admission nursing scores in the intervention group is noteworthy, especially considering the initial gap between the groups. While the comparison group showed modest improvements, the intervention group demonstrated substantial gains. This enhanced performance may be attributed to the program’s emphasis on practical exercises and real-time problem-solving scenarios [29], and our results further support this finding. Similar findings have been reported in studies examining the effectiveness of hands-on training approaches in nursing education [30].

The most dramatic improvements were observed in operation/procedure nursing and department transfer/ward transfer/discharge nursing in the intervention group. These areas, which initially showed the lowest scores, demonstrated the greatest improvement following the PEEP. This significant gain may be attributed to the program’s structured approach, which provided focused practice opportunities that are typically limited in routine clinical work [31]. The improvement in these complex tasks supports previous research indicating that workflow-centered EMR learning, combined with mentorship from experienced staff, enhances practice adaptation [28].

The improvement in competencies related to night duty responsibilities and SBAR-based handover communication warrants further examination. The intervention group showed a marked improvement in SBAR and handover skills, significantly exceeding the comparison group’s gains. This improvement is particularly important, as effective handover communication directly impacts patient safety and care continuity [32]. The structured design of the PEEP, which included specific modules on standardized communication protocols, appears to have effectively enhanced these critical skills.

The overall effectiveness of the PEEP can be attributed to several key design features. First, the integration of theoretical knowledge with extensive hands-on practice allowed participants to immediately apply the concepts they learned [29]. Second, the small-group format enabled individualized attention and immediate feedback. Third, the department-specific adaptations of the training materials enhanced the relevance and applicability of the learning content [26].

Despite their shorter average clinical experience, the intervention group demonstrated significantly greater improvements than the comparison group. While both groups showed some advancement in EMR competencies, likely reflecting natural skill acquisition through clinical exposure [30], the intervention group’s superior gains suggest that systematic education can accelerate skill development beyond what is achieved through routine clinical exposure alone. This finding has important implications for new nurse training programs, as the early implementation of structured EMR education could potentially shorten adaptation time and reduce associated stress [31].

Despite its contributions, this study has certain limitations that should be noted. First, while our sample size of 54 participants achieved adequate statistical power (0.82), it may limit the generalizability of the findings to broader nursing populations. The single-institution setting and the specific EMR system used also constrain the generalizability of our results, as different hospitals may use various EMR systems with different features and workflows. Second, while one of our study objectives was to examine the impact on turnover intention, we were unable to directly measure this outcome due to the short duration of our study design. Third, the potential Hawthorne effect must be considered, as participants were aware of their involvement in an educational intervention. Although we implemented various controls, including blinded assessment, this effect cannot be completely eliminated. Fourth, our study measured immediate and short-term outcomes, leaving questions about the long-term impact of the program on clinical performance, patient outcomes, and nurse retention. Additionally, the findings may not be fully generalizable to other hospital settings with different organizational cultures, EMR systems, or nursing practice patterns.

Based on our findings and identified limitations, we propose several recommendations for nursing education and practice. First, nursing curricula should incorporate comprehensive EMR training that includes hands-on practice with common EMR systems, workflow-based scenarios that reflect real clinical situations, and standardized documentation practices for various clinical procedures. This integration would help reduce the gap between academic preparation and the demands of clinical practice [9]. Second, healthcare institutions should consider implementing structured EMR training programs early in new nurses’ careers, with content tailored to specific departmental needs while maintaining standardized core competencies. Regular refresher training opportunities should be provided to reinforce learning and address emerging challenges.

For EMR system designers and administrators, our findings suggest the need for more intuitive user interfaces, particularly for complex procedures such as operation/procedure documentation and transfer/discharge processes. The development of context-sensitive help features and standardized templates for common nursing tasks could further support new nurses in their EMR utilization [19]. Additionally, the system should better facilitate interdepartmental communication and handovers, as these areas showed substantial room for improvement in our study.

Future research should address several key areas. Longitudinal studies are needed to evaluate the long-term impact of EMR training on clinical performance and nurse retention. Multi-center trials would enhance the generalizability of findings across different healthcare settings and EMR systems. Additionally, research examining the relationship between EMR competency and patient outcomes would provide valuable insights into the broader implications of EMR education.

## 5. Conclusions

The PEEP effectively enhanced new nurses’ EMR competencies across all measured domains, with particularly strong improvements in complex areas such as operation/procedure documentation and transfer/discharge processes. This structured, practice-oriented approach appears to accelerate EMR skill development beyond what is achieved through routine clinical exposure alone. To improve new nurses’ preparedness for modern healthcare environments, while potentially reducing stress and improving retention rates, we recommend developing comprehensive EMR education programs within nursing curricula, creating standardized competency assessment tools, and integrating workflow-based EMR training into new nurse orientation programs. Regular evaluation and updating of EMR training content will ensure continued relevance to evolving healthcare needs. Further research should examine the long-term impacts of such programs on clinical outcomes, patient safety, and professional development.

## Figures and Tables

**Figure 1 healthcare-13-00365-f001:**
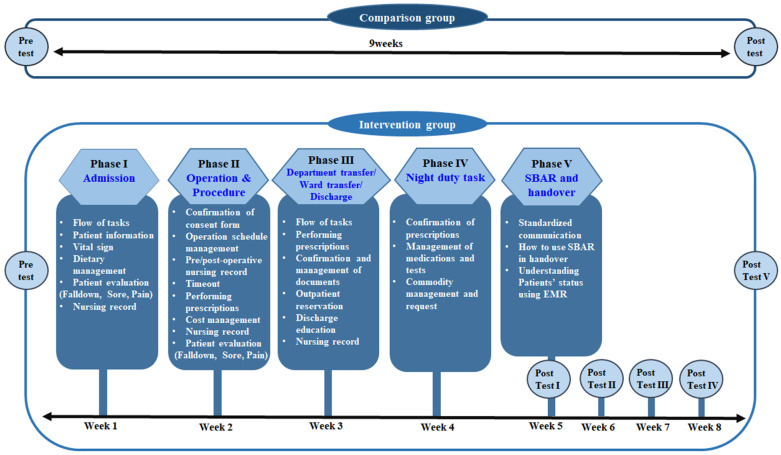
Study design.

**Table 1 healthcare-13-00365-t001:** Comparison of baseline characteristics between the two groups (N = 54).

Variables		Total	Intervention (n = 25)	Comparison (n = 29)	χ^2^/t/Z	*p*
		M ± SD OR n (%)	M ± SD OR n (%)	M ± SD OR n (%)		
Age (years)		23.81 ± 1.32	23.68 ± 1.46	23.93 ± 1.19	−1.28	0.199 ^§^
Gender	Male	6 (11.1)	5 (9.3)	1 (1.9)	3.72	0.085 ^†^
	Female	48 (88.9)	20 (37.0)	28 (51.9)		
Department	Medical ward	15 (27.8)	10 (38.5)	18 (81.8)	6.61	0.085
	Surgical ward	11 (20.4)	3 (5.6)	8 (14.8)		
	ICU	12 (22.2)	5 (9.3)	7 (13.0)		
	Others (ER, NICU)	16 (29.6)	6 (11.1)	10 (18.5)		
Working career (months)		6.09 ± 2.90	4.64 ± 2.68	7.34 ± 2.50	−3.60	<0.001 ^§^
Admission		4.09 ± 0.75	3.84 ± 0.76	4.31 ± 0.69	−2.36	0.022
Operation/procedure		3.35 ± 1.07	3.07 ± 0.83	3.58 ± 1.21	−1.82	0.075
Department transfer/ Ward transfer/Discharge		3.47 ± 0.97	3.28 ± 0.96	3.64 ± 0.97	−1.37	0.175
Night duty tasks		4.11 ± 0.78	4.05 ± 0.87	4.17 ± 0.70	−0.27	0.787 ^§^
SBAR and handover		3.85 ± 0.88	3.55 ± 0.94	4.11 ± 0.75	−2.21	0.027 ^§^

M = mean; SD = standard deviation; ICU = intensive care unit; OR = operating room; ER = emergency room; NICU = neonatal intensive care unit; SBAR = situation, background, assessment, recommendation. ^§^ Mann–Whitney U test; ^†^ Fisher’s exact test.

**Table 2 healthcare-13-00365-t002:** Comparison of the mean scores for variables between the intervention and comparison groups (N = 54).

Variables	Intervention Group (n = 25)				Comparison Group (n = 29)				Int. vs. Com.	
	Pre-Test (M ± SD)	Post-Test (M ± SD)	Difference (M ± SD)	t/z (*p*)	Pre-Test (M ± SD)	Post-Test (M ± SD)	Difference (M ± SD)	t/z (*p*)	Post-Test Only F/z (*p*)	Pre–Post Difference t/z (*p*)
Admission	3.84 ± 0.76	4.86 ± 0.29	−1.01 ± 0.72	−7.02 (<0.001)	4.31 ± 0.69	4.51 ± 0.71	−0.20 ± 0.46	−2.35 (0.026)	−18.14 (<0.001) ^†^	−4.34 (<0.001) ^‡^
Operation/ procedure	3.07 ± 0.83	4.67 ± 0.57	−1.60 ± 0.67	−11.97 (<0.001)	3.58 ± 1.21	4.09 ± 1.04	−0.51 ± 0.74	−3.72 (0.001)	−2.83 (0.005) ^‡^	−5.66 (<0.001)
Department transfer/ Ward transfer/Discharge	3.28 ± 0.96	4.75 ± 0.37	−1.47 ± 0.87	−8.49 (<0.001)	3.64 ± 0.97	4.14 ± 0.87	−0.50 ± 0.57	−4.75 (<0.001)	−3.32 (0.001) ^‡^	−4.13 (<0.001) ^‡^
Night duty tasks	4.05 ± 0.87	4.85 ± 0.31	−0.80 ± 0.77	−4.20 (<0.001) ^§^	4.17 ± 0.70	4.37 ± 0.80	−0.21 ± 0.55	−2.28 (0.023) ^§^	−3.30 (0.001) ^‡^	−2.68 (0.007) ^‡^
SBAR and handover	3.55 ± 0.94	4.77 ± 0.42	−1.23 ± 0.99	−4.29 (<0.001) ^§^	4.11 ± 0.75	4.44 ± 0.72	−0.33 ± 0.42	−3.44 (0.001) ^§^	12.31 (<0.001) ^†^	−3.25 (0.001) ^‡^

Int. = intervention group; Com. = comparison group; M = mean; SD = standard deviation. ^§^ Wilcoxon signed-rank test. ^†^ ANCOVA with pre-test score as covariate. ^‡^ Mann–Whitney U test.

## Data Availability

The data supporting the findings of this study are available from the corresponding author upon reasonable request and with permission.

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
