# Peer review of "Effect of a Practice-Oriented Electronic Medical Record Education Program for New Nurses"

_healthcare, 2025, doi:10.3390/healthcare13040365_

Round 1
Reviewer 1 Report
Comments and Suggestions for Authors
Mastering EMR systems is very important for patient care, but can also be very challenging for new nurses. Hence in this paper, the authors evaluated the effectiveness of a practice-oriented EMR education program in improving new nurses’ ability on using EMR systems. A two-group design was used, where the intervention group underwent a EMR related training and the control group did not. Then some statistical tests were conducted to evaluate the difference between these two groups.
Overall, I think this paper is written very clearly. And statistical analyses were conducted fairly comprehensively.
I just have one concern about the design. Considering that the pair t-test was used extensively in this study, and the baseline value(namely, the pre-score) is likely to be a non-negligible influencing factor, usually it is recommended to make the pre-score of the two groups ‘balanced’ so that the contribution of this factor on the comparison results can be excluded. However, as the authors have already noticed, out of the 5 dimensions, 2 of them have significant differences between the treatment group and the control group. I am wondering if it is due to some practical difficulty or if some efforts have been made to avoid this issue. For example, have the authors considered using a randomization system to assign participants (nurses) to the groups randomly? This way, the differences between the two groups on pre-scores could be reduced.
Author Response
The response to the reviewers' comments has been attached as a file.

Reviewer 2 Report
Comments and Suggestions for Authors
This study aimed to evaluate the effectiveness of a practice-oriented EMR education program designed to improve new nurses' EMR practical abilities, clinical adaptation, and reduce turnover intention. Also, this study uses the method that a quasi-experimental pretest-posttest design with a non-equivalent control group was conducted.
It is a manuscript with well-constructed and well-written content.
The references section, which I see as a significant deficiency, should be increased by adding up-to-date articles on the subject.
Comments on the Quality of English LanguageThe English could be improved to more clearly express the research.
Author Response

(The authors gave the same response as above.)

Reviewer 3 Report
Comments and Suggestions for Authors
Strengths:
- The study explicitly states its objectives: to evaluate the effectiveness of an EMR education program on new nurses' EMR abilities, clinical adaptation, and turnover intention.
- The focus on EMR proficiency for new nurses is highly relevant given the increasing reliance on technology in healthcare.
- The use of a quasi-experimental design with a control group adds strength to the study by allowing for comparison and reducing the risk of confounding factors.
- The program covered a wide range of essential EMR tasks, suggesting a comprehensive approach to enhancing new nurse skills.
- The findings have significant practical implications for nursing education, suggesting a need for more targeted EMR training programs to improve new nurse preparedness and retention.
Weaknesses:
- The sample size of 54 participants may be considered relatively small, potentially limiting the generalizability of the findings.
- The authors do not specify how turnover intention was measured or whether there were significant differences between groups in this regard. This omission weakens the study's conclusions regarding the impact of the program on nurse retention.
- As the intervention group received specialized training, a Hawthorne effect (participants modifying their behavior due to the awareness of being observed) could have influenced the results.
- The findings may not be fully generalizable to other hospital settings, different EMR systems, or nurses with varying levels of prior experience.
- The study only assessed the short-term impact of the program. Further research is needed to determine the long-term effects on nurse performance, patient outcomes, and job satisfaction.
Recommendations:
- Increase Sample Size: Conduct a larger study with a larger sample size to enhance the generalizability of the findings.
- Include Detailed Turnover Intention Data: Collect and analyze data on turnover intention using validated instruments to assess the program's impact on nurse retention.
- Control for Potential Bias: Employ strategies to minimize potential bias, such as blinding assessors to group assignment.
- Investigate Long-term Outcomes: Conduct follow-up studies to assess the long-term impact of the program on nurse performance, patient outcomes, and job satisfaction.
- Explore Program Adaptability: Investigate the feasibility and effectiveness of adapting the program to different hospital settings, EMR systems, and nurse populations.
Author Response

(The authors gave the same response as above.)

Reviewer 4 Report
Comments and Suggestions for Authors
The authors present the effect of educational program on EMR for new nurses. Introduction, materials, and methods including study design followed by the result and discussion are presented. Some points need to be addressed to improve the paper.
1. In lines 93-94, the authors state that “A Number of studies have been conducted to improve nursing practical skills using EMR through education, but they mainly targeted at nursing students and faculties”. The authors need to include a detailed literature review discussing state-of-the-art methods with the strengths and limitations of each method.
2. The contribution of the study is limited. It is a fact that after training sessions the users generally show improvement in terms of usability. The authors need to clarify the contribution.
3. It is not clear which questions were included in the study. The questionnaires used in the study need to be available as supplementary material.
4. The nurses involved in the training worked in their respective departments for five weeks alongside the training. They may have learned from the experience as well. Was this effect considered while formulating the results?
5. Based on their findings, in the discussion, the authors need to:
· Recommend the content that needs to be included in the curriculum of nursing programs. This will be helpful to avoid any additional training.
· Are there any difficulties in system usability for users? What are the recommendations for the designers of EMR?
Author Response

(The authors gave the same response as above.)

Round 2
Reviewer 4 Report
Comments and Suggestions for Authors
The authors have addressed my concerns in the previous version.
A comprehensive literature review is included in the revised version. Contribution is clarified. Supplementary material is provided as suggested and the discussion is improved as well.